# Meanings and Practices of Preceptorship in Pediatric Nursing and Their Implications for Public Health: A Grounded Theory Study

**DOI:** 10.3390/ijerph22081255

**Published:** 2025-08-11

**Authors:** Thiago Privado da Silva, Flávia Souza Soares, Italo Rodolfo Silva, Sabrina da Costa Machado Duarte, Laura Johanson da Silva, Jessica Renata Bastos Depianti

**Affiliations:** 1Nursing Institute, Multidisciplinary Center, Federal University of Rio de Janeiro, Macaé 27930-560, Brazil; italo@macae.ufrj.br; 2Anna Nery School of Nursing, Federal University of Rio de Janeiro, Rio de Janeiro 20211-110, Brazil; flavia.ssoares@fiocruz.br (F.S.S.); sabrinaduarte@eean.ufrj.br (S.d.C.M.D.); 3Alfredo Pinto School of Nursing, Federal University of State of Rio de Janeiro, Rio de Janeiro 22290-180, Brazil; laura.silva@unirio.br; 4Nursing Departament, State University of Rio de Janeiro, Rio de Janeiro 20550-900, Brazil; jessica.depianti@uerj.br

**Keywords:** nursing, pediatric nursing, preceptorship, nursing residency, public health

## Abstract

Strengthening the education of health professionals is imperative to effectively address contemporary public health challenges. Preceptorship, by integrating teaching and care within service settings, stands out as a relevant strategy for developing clinical, ethical, and relational competencies. This study aimed to construct a theoretical model based on the meanings attributed by nurse preceptors to preceptorship in pediatric nursing within the context of hospital-based training at a referral institute specializing in rare and complex diseases in Rio de Janeiro, Brazil. The study used Grounded Theory and Symbolic Interactionism as its methodological and theoretical frameworks, respectively, and involved interviews with 14 preceptors. The resulting model characterizes preceptorship as an interactive process materialized in pedagogical practices that integrate technical skill, empathy, responsibility, and creativity into the daily routine of care. The findings offer valuable insights for strengthening professional training programs in health and contribute to public policies that recognize preceptorship as a component of interprofessional education and of workforce development, with a focus on humanization, safety, and contextualized care.

## 1. Introduction

The expanding global production of health-related knowledge, in conjunction with contemporary social transformations, has intensified the international discourse on the urgent need to restructure the educational processes of health professionals. This restructuring aims to better equip them to navigate the complex challenges currently confronting health systems. Publications by the World Health Organization, the Pan American Health Organization, and the International Council of Nurses underscore this imperative by proposing educational frameworks that are evidence based, responsive to population needs, and oriented toward equity, sustainability, and the resilience of healthcare services. These strategies advocate for structural reforms in training methodologies and workforce governance, recognizing education as a foundational pillar in the transformation of health systems and the promotion of universal, high-quality access to care [1,2,3,4,5].

In Brazil, specialization programs structured as health residencies have been established as a strategy for training professionals with strong technical competence and social commitment, particularly within the Unified Health System (SUS) [6,7]. Operating under a full-time model, these programs allocate 80% of the workload to supervised practice in real healthcare settings, guided by preceptors. They represent a key phase for the development of specific competencies, especially given the generalist profile of undergraduate programs, as outlined by the National Curriculum Guidelines [8]. Within this framework, preceptorship emerges as a central component of in-service education, fostering the integration of theoretical knowledge with practical experience in authentic care environments. In this study, preceptorship is conceptualized as an educational model in which the preceptor facilitates the development of students’ problem-solving capabilities, thereby preparing them for educational, therapeutic, consultative, and managerial roles [9].

The preceptor, a health professional whose teaching role is embedded in clinical practice, plays a central part in the technical, ethical, and relational development of residents, directly influencing care quality [10]. Despite its widespread use in different countries and settings, preceptorship still lacks a clear conceptual definition, consistent institutional support, and educational models adapted to the specificities of training environments [11,12].

Pediatric nursing, the empirical focus of this study, presents unique challenges to professional education due to the complexity of caring for children and their families. This context demands refined clinical, communicative, and ethical competencies. Hospital-based practice must aim to minimize trauma, respect the stages of child development, and provide welcoming and humanized care, recognizing the child as a developing and vulnerable individual with specific needs requiring continuous, sensitive, and comprehensive attention [13].

The training of preceptors is recognized as essential, as its absence may adversely affect the socialization of newly graduated nurses within clinical environments. Moreover, the existing literature highlights the excessive workload borne by preceptors, who are tasked with simultaneously mentoring novice professionals and attending to patient care [14,15]. In this context, understanding how health professionals experience and interpret the practice of preceptorship is vital for the advancement of educational models that are grounded in service realities and aligned with population needs.

This study aimed to construct a theoretical model based on the meanings attributed by nurse preceptors to preceptorship in pediatric nursing within the context of hospital-based training at a referral institute specializing in rare and complex diseases in Rio de Janeiro, Brazil. The proposed model is intended to enhance preceptorship practices across diverse care settings by offering a structured framework to guide preceptors in shaping pedagogical processes within the caregiving domain. Although centered on nursing, the study’s findings extend to broader clinical education contexts, contributing meaningfully to pedagogical strategies across various health disciplines.

These contributions align directly with the Sustainable Development Goals of the United Nations 2030 Agenda [16], particularly with SDG 3 (ensure healthy lives and promote well-being for all at all ages), by enhancing clinical and educational practices related to child care; with SDG 4 (ensure inclusive, equitable, and quality education), by advancing educational models in the health sector; and with SDG 10 (reduce inequality within and among countries), by promoting the training of professionals committed to equity and to addressing the genuine needs of the population.

## 2. Materials and Methods

This is a qualitative study grounded in the principles of Grounded Theory (GT), as proposed by Corbin and Strauss, which enables the construction of theory through the systematic analysis and constant comparative coding of data [17]. Symbolic Interactionism was adopted as the theoretical framework to understand the meanings attributed by participants to their actions and social interactions [18]. The Consolidated Criteria for Reporting Qualitative Research (COREQ) tool was employed to ensure methodological rigor [19].

The study was conducted between November 2024 and January 2025 with pediatric nursing preceptors working in inpatient units of a public institution in the city of Rio de Janeiro, Brazil, recognized for its care of children and adolescents with rare and complex genetic diseases. The units included a Clinical Area (Pediatric Infectious Diseases, IMCU-Intermediate Care Unit, Pediatric Unit and Critical Care Unit—CCU) and Surgical Area (Surgical Neonatal Intensive Care Unit—Surgical NICU and Pediatric Surgery Unit), where preceptors supervise residents during 12 h daytime shifts.

Participants were invited in person, within the study setting. Selection was carried out through theoretical sampling, aiming to deepen the understanding of the phenomenon. The inclusion criteria comprised nurses with a specialization in child health or at least three years of experience in pediatrics and a minimum of six months of experience as preceptors. Nurses on leave during the study period were excluded. There were no refusals or withdrawals from any participants. Data collection was concluded upon reaching theoretical saturation, at which point it was observed that the categories and their respective subcategories had achieved sufficient theoretical density to elucidate the phenomenon under investigation. That is, the newly gathered data no longer contributed to modifying the consistency and depth of the established conceptual categories [17].

Data were collected through individual semi-structured interviews, which were audio recorded and fully transcribed. Each interview lasted an average of 30 min and was conducted in a private setting. The interview guide included the following initial questions: In your view, what does it mean to be a preceptor? What are your responsibilities as a preceptor in the Pediatric Area of this Institute? Due to the method’s aim of understanding meanings that emerge during interviews, transcripts were not returned to participants for revision or supplementation.

A pre-test was not conducted, as data collection and analysis occurred simultaneously, allowing for progressive refinement of the interview guide. This iterative process ensured the adequacy of the instrument through continuous theoretical development and saturation. The transcripts were analyzed using ATLAS.ti^®^ version 25.0.1, following the stages of open, axial, and integrative coding. Initially, open coding identified preliminary codes through meticulous line-by-line reading, followed by constant comparative analysis to organize conceptual codes.

During axial coding, the conditional–consequential paradigm was employed to establish relationships between categories and subcategories, considering conditions, action–interaction strategies, and consequences [17]. In the final stage, the categories were integrated, resulting in the central category: Preceptorship in Pediatric Nursing: a complex and interactive process aimed at child care. Memos and diagrams supported the theoretical integration process, and data analysis was conducted concurrently with data collection to enhance theory development.

The study adhered to the ethical guidelines established by Resolutions No. 466/2012 and No. 580/2018 of the Brazilian National Health Council and the Declaration of Helsinki [20,21,22]. It was approved by the Research Ethics Committees of both the proposing and collaborating institutions. Participants were informed about the research objectives, confidentiality, voluntary participation, and their right to withdraw at any time. They signed the Informed Consent Form and provided authorization for the recording of interviews. To ensure anonymity, the statements were identified using the letter “P” followed by the interview number (P1, P2…).

The data obtained from the interviews are confidential due to the presence of sensitive information and were therefore not made publicly available. However, they may be accessed upon reasonable request to the corresponding author, provided that ethical guidelines and the confidentiality criteria established in the informed consent are respected.

## 3. Results

With regard to the demographic and professional characteristics of the participants, fourteen nurse preceptors took part in the study. The majority were female (13), with only one male participant, reflecting the predominance of women in the nursing profession. Ages ranged from 28 to 47 years, with a mean age of 36. The length of professional experience varied from 4 to 14 years, with an average of 9 years in nursing practice. All participants were employed under the Consolidation of Labor Laws (CLT) regime, and for 12 of them, the duration of their preceptorship at the institute corresponded to their total time in the role, indicating that most had no prior experience as preceptors before assuming their current positions.

In terms of academic qualifications, all participants held a lato sensu postgraduate degree. Among them, ten preceptors had completed a residency program, eight in the field of Pediatrics and two in Neonatology, and five participants held additional specializations. Only one participant held a master’s degree, two were currently pursuing a master’s degree, and one was enrolled in a doctoral program. Notably, only one preceptor reported having undertaken a formal training course specifically designed for the role of preceptor, underscoring a significant gap in pedagogical preparation among the majority of participants.

Regarding the distribution across work units, three preceptors worked in the IMCU, two in the Surgical NICU, two in the Pediatric Unit, two in the Pediatric Infectious Diseases sector, two in the CCU.

From the data analysis, three categories emerged: understanding the factors that influence preceptorship; practicing preceptorship; and reaping the outcomes of preceptorship. When related to one another through the conditional/consequential paradigmatic model, they gave rise to the theoretical model of the phenomenon entitled “Preceptorship in Pediatric Nursing: a complex and interactive process aimed at child care”.

### 3.1. Conditions

The category “understanding the factors that influence preceptorship” contributes to understanding the context in which preceptors carry out their practice, providing a foundation for analyzing the strategies they adopt and the outcomes of their actions. This category comprises the subcategory “responding to institutional demands.” This subcategory revealed that preceptorship is not recognized by nurses as a formal responsibility or an inherent part of their role. Instead, it is perceived as an additional requirement imposed by the institution due to the presence of residents in their work environment.

For these professionals, preceptorship is incorporated into their routines implicitly, without a formal agreement or a clear definition of a specific role distinct from their care-related duties.

“*Whether we want it or not, here, nurses end up being preceptors because of the residency programs.*”(P1)

“*Since we, automatically, by working here at the institution, are preceptors, I think there should be something more—some financial incentive or even preparatory or training courses.*”(P10)

“*And it’s something that, since preceptorship is part of the institution due to the residency programs, it ended up becoming central, you know? I was hired as a nurse, but here, being a nurse automatically means being a preceptor.*”(P9)

“*I’m a shift nurse, but as a shift nurse, the residents are here with me, so I feel that duty, that obligation to be with them. […] Because I don’t hold that position here, I don’t have that role [as a preceptor], right? I’m a shift nurse.*”(P13)

### 3.2. Action–Interaction Strategies

The category “practicing preceptorship” refers to the action–interaction strategies adopted by preceptors in response to the conditions presented in the previous category. It encompasses the practices, behaviors, and approaches employed by preceptors to address those conditions, reflecting their efforts to adapt to the institutional context while emphasizing key elements in the training of residents, based on their own perceptions and experiences.

This category comprises the subcategories “fulfilling technical functions,” “humanizing teaching and taking responsibility for the student,” and “developing their own teaching methods.” These subcategories elucidate how preceptors conduct their pedagogical activities, how they engage in the educational process of residents, and what resources and strategies they utilize in fulfilling their role.

The narratives included in the subcategory “fulfilling technical functions*”* underscore that pedagogical practices aimed at technical skill development are both comprehensive and dynamic. These practices include welcoming the resident; introducing the unit and the clinical cases of pediatric patients; conducting theoretical discussions; performing suctioning of upper and lower airways; preparing for entry into isolation rooms; executing catheterizations; administering Total Parenteral Nutrition (TPN); performing peripheral venipuncture and insertion of Peripherally Inserted Central Catheters (PICCs); conducting other invasive procedures within the nursing scope of practice; providing hygiene care; changing dressings; managing medical prescriptions; preparing medications; overseeing preoperative preparation, admission, and postoperative care; as well as coordinating and evaluating the resident’s performance.

“*I show the entire unit, explain how it works, and go over each child in detail {…} how we care for a child, how to perform suctioning. I demonstrate every type of care provided here.*”(P1)

“*We start with the theoretical part […] and then we move on to the practical part.*” (P2)

“*So, here we perform various types of procedures—we administer parenteral nutrition, we insert feeding tubes {…} I also share books I know, that I’ve read and believe will be professionally useful for them. It’s these two aspects: practice and theory.*”(P12)

“*I think being a preceptor is mainly about teaching, welcoming, and coordinating the residents who are assigned to me.*”(P3)

“*We teach them what the functions are, what they need to do, how to prepare to enter an isolation room, and how to exit. The other technical procedures too, like venipuncture, insertion of a PICC (Peripherally Inserted Central Catheter). So, I think it’s really our entire daily routine. There’s no difference between our role and theirs.*”(P5)

“*So, whenever I can, I explain how to change the dressing on a central catheter—procedures that are more invasive and specific to nurses. I also do evaluations, because I end up assessing them since I work directly with them.*”(P13)

The subcategory “humanizing teaching and taking responsibility for the student” underscores the preceptors’ commitment to delivering instruction in a humanized and empathetic manner. It reveals their dedication to continuous professional development, their proactive engagement in the educational formation of residents, and their unwavering concern for child safety. This pedagogical stance is manifested through practices that emphasize receptiveness, open dialogue, and sustained support for residents throughout their learning journey.

“*I believe it is a great responsibility because we teach care. In the best way and with a lot of responsibility. If I were to define it in one word, preceptorship is responsibility for care*”(E14).

“*I even talk about everyday experiences I’ve had elsewhere, in other hospitals. In fact, I even do a bit of psychology with them, because they’re very afraid of what it’ll be like out there. I tell them how good they are, so they don’t get stuck in that fear, you know. And whenever they face difficulties, I talk to them and point out everything they need to improve. I do that too.*”(P8)

“*Not long ago, a resident—I’ll even use this as an example—told me she had a lot of trouble with venipuncture. So I created a little practice setup using a small tube to help her get the angle right. I drew it out, made it look like a vein, taped a tube on top so she could figure out the exact angle the needle should go in. She practiced there before puncturing the child. That made her feel more confident.*”(P11)

“*I always try to be present for any kind of procedure they’re going to perform. I know it’s not always possible because there’s a lot going on, but I try. They might have a question in the moment, or maybe they don’t know something, or they’re too shy to ask. So I always try to be there to make sure no incidents happen. I always make myself available if they need me.*”(P9)

“*So, as preceptors, we need to be prepared to answer the residents’ questions. We need to study to stay up to date, whether it’s about procedures or the rare pathologies we see here. We always have to be studying and updating ourselves in order to teach and educate someone.*”(P6)

The subcategory “developing their own teaching methods” reveals the strategies preceptors use to teach. They create resources to support learning amid high work demands.

“*Each of us ends up having our own method. I have mine—I leave them with a nursing technician to learn the more basic care, and then I spend time with them whenever I can. Gradually, I let them go and start assigning them to a more stable patient at first, then gradually to a more critical one.*”(P13)

“*I go into the room and explain things as I do them, because I’m the shift nurse and often don’t have time to take them aside. I also don’t have time to supervise them properly, because I have to manage the shift and keep an eye on everything.*”(P13)

“*I base myself on my professional experience—and also on having been a resident. Since we’ve been residents, we have a sense of what it’s like to be a preceptor. I relied on many professionals, on preceptors who passed on their experience to me. So much of what I learned came from preceptors and from hands-on professionals. I observed what they did, and today I kind of replicate what I learned. It was life experience.*”(P10)

“*I always say that the one who taught me wasn’t a nurse, it was a nursing technician. Because the nurses where I trained didn’t have the patience or willingness to teach. So I try to do things differently from what I went through.*”(P1)

“*Since I’ve worked in several units here, we end up gaining a bit of experience, right. And I’ve also worked in other hospitals, including emergency care, so we bring some of that practical background with us. From experience. But I think there should be more training opportunities, more preparatory courses—that would be really helpful.*”(P11)

### 3.3. Consequences

The practice of preceptorship in pediatric nursing residency emerges as a dynamic and relational process, wherein preceptors not only teach and accompany residents but also experience the personal and professional repercussions of this role. Within this context, the “*consequences*” element of the paradigmatic model becomes evident, highlighting the outcomes of preceptorship from the perspective of the preceptors themselves. This gives rise to the category “reaping the outcomes of preceptorship,” which comprises two subcategories: “contributing to quality training” and “feeling overwhelmed.”.

The subcategory “contributing to quality training” captures preceptors’ reflections on the impact of their role in shaping the educational trajectory of pediatric nursing residents. Despite the challenges encountered during residency, preceptors underscore that clinical practice progressively cultivates the development of essential competencies for pediatric care. They emphasize that their contribution transcends technical instruction, encompassing ethical formation and the promotion of critical thinking.

Moreover, preceptors recognize their influence on residents’ attitudes, values, and professional outlooks, potentially fostering a more reflective and socially committed approach to nursing. Nonetheless, they caution that when this influence is inadequately exercised, it may inadvertently impede the residents’ professional development.

“*The residents who come through the institute leave very well prepared. They face challenges along the way, just like we do as preceptors, but they leave here very well prepared.*”(P2)

“*You can see how unprepared they are when they graduate, and how they develop in the field as they interact with the population we serve here. As they ask questions and learn from the preceptors.*”(P6)

“*I think we’re able to train highly qualified nurses for the workplace. We know that some people have some experience or some kind of specialization, but not everyone has this level of knowledge.*” (P12)

“*We’re able to support them, as preceptors, in their journey toward excellence.*”(P14)

“*We have a lot of influence on how students form their opinions and perspectives, on their ability to think critically, you know? To feel responsible for the patient. To feel like a member of the care team while they’re working with us. We shape a lot of opinions. That can be a very good thing. But it can also be a vulnerability, right?*”(P7)

The subcategory “feeling overwhelmed” exposes the adverse effects stemming from the meanings, efforts, and strategies adopted by preceptors, particularly when juxtaposed with their aspiration to deliver high-quality training. While they recognize the value of their contributions to the educational process, preceptors also report experiencing fatigue and a sense of overload.

This subcategory underscores the emotional and professional toll associated with the continuous endeavor to meet pedagogical expectations, especially when these demands are intensified by clinical responsibilities and institutional constraints. Within this context, preceptors express a sense of frustration, perceiving that, despite their commitment and dedication, they fall short of providing the ideal preceptorship they aim to offer.

“*I see proper preceptorship as following the resident through all the activities they perform, teaching how things should be done and answering questions*”(P9)

“*I can’t give them all the time they would need to learn. I just can’t. I have to split my time between them and my other tasks*”(P3)

“*I work in an ICU where I have to handle everything, and sometimes I have two residents*”(P8)

“*So you need to be willing to teach when there are questions, to share your knowledge—whether it’s clinical or bureaucratic—and really pass on everything you know to someone else in a calm way. I think that’s it. Being available for that, being a reference for those people […] I think this work overload, having so much to do, ends up reducing the time we could dedicate to sharing that knowledge with them*”(P12)

Considering the above, Figure 1 presents the diagram representing the theoretical model of the phenomenon that emerged from an in-depth analysis of the data: “Preceptorship in Pediatric Nursing: a complex and interactive process aimed at child care.”

The model expresses the meaning of preceptorship as constructed by preceptors based on social interactions within the pediatric inpatient units of the institute. Preceptorship is understood as a multifaceted activity and is evidenced as a practice formed through the articulation of knowledge, experience, and interpersonal relationships, with qualified child care as its central axis.

## 4. Discussion

The theoretical model was developed based on the core elements of Grounded Theory, namely conditions, action–interaction strategies, and consequences [17]. It represents how the meaning of preceptorship in pediatric nursing residency is continuously constructed and reconstructed through interactions with the institution, healthcare professionals, residents, and children. These meanings emerge from social relations, guide human behavior, and are constantly reshaped through interpretive processes [18].

In the category “understanding the factors influencing preceptorship,” the subcategory “meeting the demands of the institutional context” emphasizes the motivations that lead nurses to undertake the role of preceptors, as well as the moment when this responsibility becomes embedded within their professional routine. From the Symbolic Interactionism perspective [18], the meanings attributed to preceptorship are produced through daily interactions with colleagues, residents, and the institution itself. Consequently, this practice is often perceived less as a discrete formal assignment and more as an inherent demand of the work environment.

This perception arises from the simultaneous identification of the residents’ needs and institutional expectations. The commitment to teaching emerges from daily practice and the aim to ensure high-quality care for children, grounded more in a collective sense of responsibility than in explicit institutional recognition.

Even in the absence of formal designation as educators, preceptors remain actively engaged, motivated by the understanding that residents’ learning is integral to the continuity and safety of patient care. This engagement reflects an internalized social meaning [18], which informs their pedagogical actions despite the lack of formal institutional support.

Supporting this analysis, the existing literature affirms that preceptors acknowledge the significance of their role in the teaching–learning process, viewing it as a meaningful contribution to the professional development of residents, even when their position lacks formal definition [8]. Accordingly, preceptors’ motivation is closely linked to their recognition of the importance of student learning in maintaining the quality of care [23].

While the individual willingness of preceptors is relevant, studies warn that without institutional support (such as reduced workload, pedagogical training, and formal recognition of the role) this willingness alone does not ensure effective teaching or the development of professionals with competencies aligned with current healthcare demands [14,24,25]. There is a pressing need to consolidate preceptorship as a shared responsibility among policymakers, institutions, and healthcare professionals, supported by investments in structural conditions that uphold it as a legitimate educational practice guided by the needs of individuals and families [26,27]

Given this context, “practicing preceptorship” emerges as the second element of the model: a set of action and interaction strategies employed in the teaching of medical residents. This category reflects how preceptors integrate technical skills, human values, a sense of responsibility, and creativity within the framework of pediatric care and education, thereby revealing the meanings ascribed to preceptorship through interactions with residents, healthcare teams, and the institutional environment.

While fulfilling technical responsibilities within the scope of preceptorship, preceptors also engage in pedagogical activities aimed at instructing residents in clinical procedures such as venipuncture, wound dressing, and medication preparation, among other techniques specific to pediatric nursing. Through their investment in the training and refinement of practical skills, these professionals contribute not only to safer and more effective patient care but also to the overall enhancement of healthcare service quality [28].

The actions undertaken reveal that although primarily technical, they are imbued with symbolic meaning and reflect the preceptor’s proactive commitment to sharing their knowledge through a deliberate pedagogical approach. This instructional approach regards technical education as a continuous process embedded within routine clinical practice, transcending the mere conveyance of procedural information. The act of welcoming residents—introducing them to the unit, its protocols, and operational routines—constitutes an intentional effort to integrate learners into the collective environment and to provide a sense of security at the outset of their formative journey.

Symbolically, the preceptor constructs a social space in which the resident is able to exist, learn, and grow, thereby legitimizing their presence in the care setting. From the perspective of Symbolic Interactionism [18], this can be interpreted as a form of subjective validation that arises through interpersonal interactions. Student evaluations conducted by preceptors can also be understood through the lens of Symbolic Interactionism, serving as a communicative act that conveys expectations, recognition, and interpretations formed through everyday exchanges. In this context, assessment becomes a valuable tool that helps students reflect on their practice, redirect their approach, and adjust their learning trajectory as needed [29].

The establishment of bonds between preceptors and residents is essential for fostering a collaborative and effective learning environment [30,31]. However, for such investment to yield substantive outcomes in resident training, the transmission of technical knowledge must be complemented by pedagogical strategies that promote not only procedural proficiency but also the development of critical thinking and collaborative competencies essential to patient care [32].

The commitment to continuous knowledge updating reflects the preceptors’ dedication to evidence-based teaching. This attitude is consistent with the findings of a study [2], which indicates that preceptors who are aware of the significance of their role tend to be better prepared, actively pursue ongoing professional development, and engage more deeply in the teaching–learning process. Nevertheless, this pursuit of continual improvement unfolds within a context characterized by the absence of a formal, structured, and institutionalized teaching model.

From the perspective of Symbolic Interactionism [18], when preceptors engage in “humanizing instruction and assuming responsibility for the student,” they are constructing meaning through their interactions with residents. The teaching of care is thus tailored to the individual needs of each student, with an emphasis on quality of assistance and professional development. The use of dialogue and empathy in addressing expressed doubts and anxieties reveals a sustained effort to interpret and respond to the symbols manifested by learners. This approach fosters pedagogical practices that are attuned to each participant’s individuality and promotes a welcoming educational environment.

The literature highlights that effective communication, emotional support, and the establishment of trust between preceptors and residents are essential for cultivating a safe and productive learning environment. These elements not only enhance the ability to confront challenges and foster the development of professional confidence but also directly influence students’ capacity to cultivate empathy and improve the provision of humanized, high-quality care [33,34].

Close supervision, valued by preceptors as a means of ensuring both child safety and teaching quality, also aligns with the principles of Symbolic Interactionism [18], which acknowledges that knowledge is gradually constructed through experience and continuous dialogue. Within this framework, errors are not perceived as isolated failures, but rather as integral components of a symbolic process of knowledge construction, wherein each action or reaction is interpreted contextually.

Ultimately, the commitment to continuous knowledge updating reflects preceptors’ understanding of their social and formative role. This attitude aligns with the findings of a study [27], which indicates that preceptors who are aware of the importance of their function tend to be better prepared, pursue ongoing professional development, and engage more actively in the teaching–learning process. This pursuit of improvement may be interpreted as an expression of professional identity, continually negotiated in relation to the values of educational practice and shaped through workplace interactions, as proposed by Blumer [18] in his theory of the continuous interpretation and negotiation of meaning in social interactions. However, such updating occurs within a context marked by the absence of a formal and institutionalized teaching model for the sharing of meaning, requiring preceptors to symbolically construct their own educational practices, guided by values and meanings formed through everyday interactions.

In this context, the subcategory “developing individualized teaching methods” illustrates how preceptors employ distinctive approaches to guide residents’ learning. These are context-specific strategies that emerge from everyday experience and the imperative to adapt the teaching–learning process to the realities of clinical practice. From Blumer’s [18] perspective, this dynamic characterizes preceptorship as an interactive and symbolic practice in which the preceptor continuously interprets the clinical environment, the residents’ behaviors, and the institutional demands. As a result of this interpretive process, teaching acquires a personal and creative dimension, manifesting as a pedagogical practice rooted in lived experience, close observation, and constant reflection on the meaning of teaching within the context of pediatric hospitalizations at the institute.

Preceptors demonstrate an individual commitment to transforming routine care into learning opportunities. They position themselves as role models and use hands-on demonstration combined with explanation as a pedagogical tool, which also helps optimize the available time. This method, however, is also perceived as challenging, as it requires a constant attitude of reflection and awareness of their role as educators. In seeking references to guide their teaching practices, some preceptors adopt institutional clinical protocols. This choice underscores the influence of institutional norms on the conduct of teaching within the clinical setting.

Clinical protocols function as shared symbolic references among healthcare professionals, serving as practical guides and shaping the residents’ learning process through standardized actions grounded in collective experience. This suggests that residents’ learning is mediated by socially constructed values and practices, reinforcing the symbolic and interactional dimension of preceptorship, in accordance with the principles of Symbolic Interactionism, which emphasize the social construction of meaning through shared symbols and interaction [18]. Preceptors also face the paradox of wanting to engage in the training of residents while work demands threaten this mission. As a result, they develop strategies to support residents’ learning and ensure child safety.

When direct supervision is not feasible, preceptors delegate part of the teaching process to other professionals, such as nursing technicians and routine nurses, who typically hold distinct responsibilities. In Brazilian hospital practice, nursing technicians are mid-level professionals with technical training who provide direct patient care under the supervision of registered nurses, who are university graduates with broader responsibilities [35].

Routine nurses are responsible for managing duty schedules, supervising the nursing team, performing administrative tasks, and ensuring compliance with safety and quality standards. In contrast, clinical nurses are directly responsible for patient care, supervision of nursing technicians, and the implementation of nursing interventions based on institutional protocols and the nursing process [35].

Although both roles are governed by the same legislation (Law No. 7498 [36]), their responsibilities differ in focus: direct patient care (clinical nurses) versus coordination and quality assurance (staff nurses). In the context studied, clinical nurses are directly responsible for supervising residents during their practice. The strategy of delegating teaching responsibilities reveals that teaching occurs in a decentralized manner and relies on team collaboration.

These findings reinforce the existence of a demand for collaborative participation from other professionals in the educational process, which is essential for strengthening teaching and learning, as well as for fostering interaction among team members, as noted in the literature [32]. Teaching is configured as a dynamic and interactive process, shaped by personal experiences and the relationships established in the exercise of the role.

Individual trajectories directly influence the construction of the meanings attributed to being a preceptor. It is common for previous experiences (such as their time as residents or their clinical practice in pediatric care) to serve as references for developing their own teaching strategies. This demonstrates that pediatric nursing preceptorship constitutes an interpretive process in which preceptors internalize patterns of conduct based on past interactions and reframe them for current practice, in line with the assumptions of Symbolic Interactionism [18].

The development of individual teaching methods does not occur in isolation. Rather, it reflects the absence of institutional guidelines that provide pedagogical support to preceptors. Previous studies show that the lack of formal and structured training leads these professionals to rely on practical experience and daily observation as their main sources for guiding the educational process. The absence of clear guidance and specific preparation generates uncertainty regarding which pedagogical strategies to adopt, requiring constant adaptation to the clinical context and driving the creation of personalized approaches grounded in their lived experiences and the realities of care [30,37,38].

Through this process, preceptors begin to recognize the impact of their work, both on residents’ development and on their own professional trajectories. The category “reaping the outcomes of preceptorship” emerges from this context within the theoretical model, reflecting the meanings attributed by preceptors to the effects of preceptorship. These perceptions arise through personal interpretations formed during the social interactions occurring within the institution, particularly in the bonds established with residents, children, and other professionals, as well as in the pedagogical efforts undertaken throughout daily practice. These meanings are understood to be continuously generated and reshaped through interaction [18]. As interpretations grounded in lived experiences within workplace relationships, the recognition of preceptorship outcomes constitutes a symbolic and interactive phenomenon.

Within this category, the subcategory “contributing to quality training” highlights preceptors’ perceptions of the positive impact of their role in residents’ professional development. They report that residents complete the program equipped for the job market, demonstrating technical proficiency and critical competencies required for clinical practice. This understanding is shaped through daily interactions and ongoing observation of the residents’ trajectories, forming an interpretive process that gives meaning to the preceptor’s role as an educator. Thus, practical experience, combined with pedagogical support, is understood as a key element in the progressive development of essential skills for pediatric care.

Preceptors observe that by the end of the program, residents exhibit technical competencies and critical thinking skills. This assessment is anchored in daily interactions that allow preceptors to recognize the residents’ development as a reflection of their own pedagogical work. Symbolic Interactionism [18] helps elucidate how these interpretations are shaped by shared symbols in the everyday preceptorship setting, including attitudes, behaviors, language, and institutional routines, which guide actions and construct shared meanings about what it means to “teach well” or “provide quality training.” This meaning-making process is also influenced by the preceptors’ personal histories, the relationships they establish with residents, and the symbolic exchanges that unfold in this context.

The literature supports these perceptions by emphasizing the pivotal role preceptors play in facilitating students’ adaptation to the professional environment and in cultivating essential clinical competencies. As educators, role models, and evaluators, preceptors provide residents with meaningful learning experiences mediated by interpersonal relationships that foster knowledge construction [14,30].

In addition, aspects such as opportunities for professional growth, the satisfaction of sharing knowledge, and the ability to contribute to the training of new professionals are perceived by preceptors as benefits of their role, strengthening their sense of commitment to preceptorship [12,23].

However, the symbolic dimension of preceptorship is also marked by tensions. One study points out that the lack of structured training for preceptors can lead to stress, particularly when combined with heavy workloads [39]. Supporting this finding, the subcategory “feeling overwhelmed” reveals the strain experienced by preceptors as they navigate the accumulation of clinical and educational responsibilities. The pursuit of high-quality education without clear guidelines demands ongoing effort, supervision, and mentorship, yet these actions are not always feasible within the constraints of clinical duties, leading to frustration and burnout.

Moreover, the concern with ensuring the child’s safety and quality of care while supervising residents imposes constant pressure and stress. The ideals of preceptorship and the desired model of comprehensive resident supervision—valued and aspired to by preceptors—are often undermined by workload demands and institutional challenges associated with clinical teaching. This results in a perceived inability to devote sufficient time and attention to residents.

This experience is also interpreted by preceptors through the lens of their professional practice and personal values, revealing a continuous process of adaptation and negotiation of meaning in the face of adversity, as proposed by Symbolic Interactionism [18]. Additional stressors stem from the effort required to balance clinical and educational responsibilities without a reduction in working hours [23,25,40]. The overlap of roles intensifies daily pressure and compromises their ability to fully carry out their educational responsibilities.

In this context, organizational and psychological support, fair remuneration, institutional recognition, and opportunities for professional development emerge as essential elements in fostering emotional resilience among preceptors in high-demand settings [23,30]. Measures such as humanized management, effective communication channels, and institutional support contribute to nurses’ well-being, mitigating the negative effects of stress and improving their ability to perform their educational roles under pressure [14,24,39].

The theoretical model proposed in this study, although grounded in the specific context of Brazilian residency programs, offers a meaningful contribution to the global discourse on clinical preceptorship. It underscores elements that transcend conventional frameworks of clinical education. Notably, the decentralization of teaching responsibilities among healthcare team members, the individualized pedagogical creativity, and the interpretive nature of the preceptor’s role constitute conceptual innovations. These align with international guidelines established by the World Health Organization (WHO) and the Pan American Health Organization (PAHO), which advocate for interprofessional education, the enhancement of teaching competencies, and the recognition of the nursing team as a central educational agent within health systems [1,2,3,4,41].

This interpretive and collective approach complements structured models which, while effective, often standardize educational practices and overlook the local context, educator subjectivity, and the complexity of clinical environments, dimensions underscored in international settings that highlight the need for adaptive approaches grounded in institutional support and pedagogical development of clinical preceptors [11,23,37].

Furthermore, the absence of family engagement in the educational process, as observed in the present study, highlights a critical gap when examined against WHO’s Global Strategy on Integrated People-Centred Health Services (2016–2026), which advocates for patient-centered care and the active participation of families and communities as pillars of humanized and transformative healthcare [41].

The findings presented herein have broadened applicability across international contexts, particularly in countries facing comparable challenges in clinical training. Strategies such as reducing the clinical workload of preceptors, establishing structured pedagogical training programs, institutional recognition of nursing’s educational role, and ensuring emotional and organizational support are consistently endorsed in the international literature as key elements for strengthening sustainable, person-centered educational practices [2,5,25,30].

While the study’s sample size and its confinement to a single institution limit the generalizability of the results and the validation process remains ongoing, the insights generated may nonetheless prove valuable for analogous contexts involving professional education and the teaching–learning process.

Future research should aim to deepen the understanding of residents, nursing technicians, and other healthcare professionals regarding the preceptorship process. Such inquiry would enhance comprehension of educational practices within collaborative frameworks, taking into account the shared meanings constructed among individuals, institutions, and public health policies. Investigating the impact of structured preceptorship on care quality and the safety of pediatric patients and their families in complex healthcare settings may yield critical evidence to inform and strengthen public policy.

Additionally, the development and evaluation of pedagogical strategies designed to improve preceptor qualifications are recommended. These efforts should be carefully aligned with institutional values and governmental directives. The study also encourages the exploration of new methodological approaches to the topic, including quantitative and mixed-method designs, which can offer a multidimensional perspective. Finally, incorporating family engagement into the educational process may contribute to the formation of professionals who are more responsive to users’ needs and committed to person-centered, equity-driven care.

## 5. Conclusions

The data analysis of this study, conducted in accordance with Grounded Theory and interpreted through the lens of Symbolic Interactionism, enabled the construction of a theoretical model that reveals the meaning attributed by preceptor nurses to the practice of preceptorship in the pediatric nursing residency of an institute in Rio de Janeiro, Brazil. This model holds significant potential to inform institutional decision making and public policy initiatives that recognize preceptorship as a foundational element of professional education in healthcare, with particular emphasis on humanized care, patient safety, and workforce development.

The theoretical model conceptualizes preceptorship as a complex and interactive process oriented toward pediatric care. It is sustained through continuous and dynamic interactions among preceptors, residents, the multiprofessional healthcare team, pediatric patients, and the institution itself. The meanings ascribed to preceptorship are shaped by these social relationships, continuously negotiated and reinterpreted, and ultimately expressed through pedagogical practices that integrate technical proficiency, empathy, accountability, and creativity into the daily routines of clinical care.

Despite the absence of formal recognition, preceptors embrace teaching as an intrinsic dimension of their caregiving responsibilities, motivated by a collective sense of professional commitment. In the absence of structured pedagogical frameworks, they devise individualized teaching strategies, frequently drawing upon institutional protocols that serve as shared symbolic references for clinical practice and in-service education.

This conceptualization of preceptorship as a socially constructed, symbolically mediated, and contextually situated practice underscores its value as a formative mechanism for enhancing the quality of pediatric healthcare. Beyond its institutional function, preceptorship emerges as a strategic instrument for reinforcing health systems that aspire to be more effective, compassionate, and equitable. Accordingly, investment in the formal recognition, pedagogical support, and institutionalization of preceptorship represents a strategic initiative for advancing public health. It directly contributes to the preparation of healthcare professionals who are committed to person-centered care and aligned with the guiding principles of Brazil’s Unified Health System (Sistema Único de Saúde—SUS).

## Figures and Tables

**Figure 1 ijerph-22-01255-f001:**
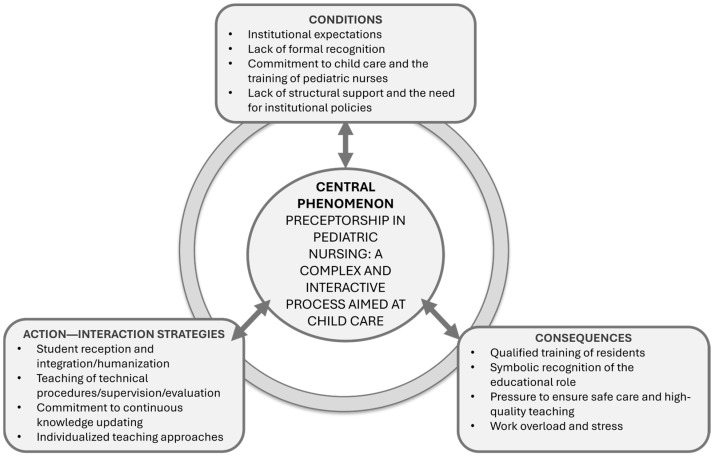
Diagram representing the theoretical model “Preceptorship in Pediatric Nursing: a complex and interactive process aimed at child care.” Source: Study data based on the paradigmatic model of Grounded Theory.

## Data Availability

The datasets generated during the study are not publicly available due to ethical and privacy restrictions but are available from the corresponding author upon reasonable request.

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
