# Peer review of "Meanings and Practices of Preceptorship in Pediatric Nursing and Their Implications for Public Health: A Grounded Theory Study"

_ijerph, 2025, doi:10.3390/ijerph22081255_

Round 1
Reviewer 1 Report
Comments and Suggestions for Authors
Abstract: The objective presented in the introduction should be clearly stated in the abstract. Please remove the content currently in lines 28 and 29 and instead present the main result of the grounded theory. Keywords: Include nursing. Introduction: Lines 76 to 78 – This content should be moved to the conclusion. Methods: How were participants recruited for the study? Did the interview guide consist of only two questions? Was a pre-test of the data collection guide conducted? Were repeated interviews carried out? If so, how many? Were the transcripts returned to participants for comments and/or corrections? Results: In my opinion, Table 1 is unnecessary, as all the information it contains is already presented in lines 135 to 152. It is assumed that tables are self-explanatory, and the authors should highlight only the most relevant findings in the text. Figure 1: The term "strategies" was used instead of "Action-Interaction Strategies," as indicated in section 3.2 of the results and also in the methods description. I suggest using the term "Action-Interaction Strategies" for consistency. Discussion: All categories and subcategories should be discussed based on the theoretical framework of symbolic interactionism. Check the paragraph length – there are some paragraphs consisting of only one sentence. Insert references for the statements in lines 350 to 353, 426 to 428, 439 to 440, and 496 to 498. Lines 432–433: What are nursing technicians and staff nurses? What are the additional responsibilities of these professionals? The study presents a grounded theory applied to the Brazilian context. However, the discussion should better emphasize how the most relevant findings significantly contribute to the existing body of knowledge in the field, from a global perspective.
Author Response
Comments: Abstract: The objective presented in the introduction should be clearly stated in the abstract. Please remove the content currently in lines 28 and 29 and instead present the main result of the grounded theory. Keywords: Include nursing.
Response: Thank you for pointing this out! We concur with your observation and have made the necessary revisions. The modifications are indicated in red for you convenience. Lines 23-26
Comments: Introduction. Lines 76 to 78 – This content should be moved to the conclusion.
Response: We agree with your observation. The modifications are indicated in red for you convenience. Lines 644-647.
Comments: How were participants recruited for the study? Did the interview guide consist of only two questions? Was a pre-test of the data collection guide conducted? Were repeated interviews carried out? If so, how many? Were the transcripts returned to participants for comments and/or corrections?
Response: We thank you for pointing this out. The responses to the questions are indicated in red for your convenience. Lines 113-115; 125-129; 117-122.
Comments: Results: In my opinion, Table 1 is unnecessary, as all the information it contains is already presented in lines 135 to 152. It is assumed that tables are self-explanatory, and the authors should highlight only the most relevant findings in the text. Figure 1: The term "strategies" was used instead of "Action-Interaction Strategies," as indicated in section 3.2 of the results and also in the methods description. I suggest using the term "Action-Interaction Strategies" for consistency.
Response: We agree with the comments. The table 1 was removed and the figure 1 was modificated as requested. Line 348
Comments: All categories and subcategories should be discussed based on the theoretical framework of symbolic interactionism. Check the paragraph length – there are some paragraphs consisting of only one sentence. Insert references for the statements in lines 350 to 353, 426 to 428, 439 to 440, and 496 to 498. Lines 432–433: What are nursing technicians and staff nurses? What are the additional responsibilities of these professionals? The study presents a grounded theory applied to the Brazilian context. However, the discussion should better emphasize how the most relevant findings significantly contribute to the existing body of knowledge in the field, from a global perspective.
Response: We thank you for pointing this out! We have made modifications based on these comments. They are indicated in red. Lines 496-505; 613-618

Reviewer 2 Report
Comments and Suggestions for Authors
1. introduction
Lack of a Clear Conceptual Definition of Preceptorship
While the introduction acknowledges that the concept of preceptorship is widely used yet inconsistently defined, the study does not sufficiently clarify how it conceptualizes preceptorship in its own framework. It is recommended to include a more explicit definition of preceptorship by comparing existing definitions and highlighting the theoretical contribution of this study.
2. Low Readability of Table 1
Table 1 presents participant characteristics in a fragmented manner, which affects its readability. Consider restructuring the table to make it more concise and reader-friendly by clearly separating variables and categories and using a more compact format.
3. Absence of the Family-Centered Care Dimension
As noted in the discussion, the absence of any mention of the child’s family in the narratives of the preceptors represents a significant gap, particularly in pediatric nursing. Including questions or reflections related to family-centered care in future data collection would align the study more closely with international guidelines for holistic, people-centered care.
4. Limited Representativeness of the Sample
This study aims to explore the approaches used by NH managers to facilitate a CLE which promotes learners’ positive learning experiences and reflects basic nursing. However, the research subjects included not only nursing managers but also nursing students and trainees, making them unsuitable for the study. While the sample size (14 participants) is adequate for a qualitative study, the fact that all participants are from a single institution may limit the generalizability of the findings. Including preceptors from multiple institutions or different healthcare settings could enrich the theoretical model and broaden its applicability. Of course, as this study is a qualitative study, it is more important to determine whether the content of the study has been saturated regardless of the number of people, but such content has not been written.
5. Lack of Quantitative or Mixed-Methods Follow-Up
The qualitative approach is well-executed, but the study could benefit from recommending follow-up research using quantitative or mixed-method designs. This would help evaluate the impact of structured preceptorship programs and validate the theoretical model in diverse settings.
6. References
There is also a problem with the references in this study, as 62% of them are more than 5 years old.
Author Response
Comments: While the introduction acknowledges that the concept of preceptorship is widely used yet inconsistently defined, the study does not sufficiently clarify how it conceptualizes preceptorship in its own framework. It is recommended to include a more explicit definition of preceptorship by comparing existing definitions and highlighting the theoretical contribution of this study.
Response:
We thank you for pointing this out. An explicit definition of preceptorship has been incorporated as requested. The theoretical contributions of this study are highlighted in red. Lines 59-62.
Comments:
Table 1 presents participant characteristics in a fragmented manner, which affects its readability. Consider restructuring the table to make it more concise and reader-friendly by clearly separating variables and categories and using a more compact format.
Response: We thank you for the recommendation. We chose to remove the table, as suggested by one of the reviewers, since all the information it contained is already presented in the first three paragraphs of the introduction. This decision was made to avoid redundancy.
Comments:
As noted in the discussion, the absence of any mention of the child’s family in the narratives of the preceptors represents a significant gap, particularly in pediatric nursing. Including questions or reflections related to family-centered care in future data collection would align the study more closely with international guidelines for holistic, people-centered care.
Response:
We thank you for point this out. This finding was unexpected and surprised the authors of the study. It was anticipated that preceptors, in terms of meaning-making, would demonstrate concern for including the family in the teaching and learning processes involved in pediatric nursing residency. However, the result reveals that the teaching and care practices within the studied context are rooted in a child-centered model. For this reason, we have added a paragraph in the discussion to suggest that future research be conducted to explore how the family might be incorporated into the professional training process of preceptors. This addition is highlighted in red. Lines 635-638.
Comments:
This study aims to explore the approaches used by NH managers to facilitate a CLE which promotes learners’ positive learning experiences and reflects basic nursing. However, the research subjects included not only nursing managers but also nursing students and trainees, making them unsuitable for the study. While the sample size (14 participants) is adequate for a qualitative study, the fact that all participants are from a single institution may limit the generalizability of the findings. Including preceptors from multiple institutions or different healthcare settings could enrich the theoretical model and broaden its applicability. Of course, as this study is a qualitative study, it is more important to determine whether the content of the study has been saturated regardless of the number of people, but such content has not been written.
Response:
We thank you for pointing this out. Indeed, the findings exhibit limitations regarding their generalizability, considering that the study was conducted with participants from a single institution. For this reason, we have decided to include a paragraph in the conclusion addressing the limitations of the results, among which is their restricted applicability to different realities. Nonetheless, it is important to emphasize that the theoretical model developed may prove useful in analogous contexts where pediatric nursing preceptorship is practiced.
We chose to make the theoretical saturation of the study more explicit within the methodology section. Lines 118-122.
Comments:
The qualitative approach is well-executed, but the study could benefit from recommending follow-up research using quantitative or mixed-method designs. This would help evaluate the impact of structured preceptorship programs and validate the theoretical model in diverse settings.
Response:
We thank you for the recommendation. We concur with the reviewer and have included a paragraph in the discussion highlighting the limitations of the study, as well as recommending future research on the topic. The modifications are indicated in red. Lines 620-637.
Comments: There is also a problem with the references in this study, as 62% of them are more than 5 years old.
Response:
We thank you for point this out. There is no mention in the journal's guidelines regarding the required percentage of references from articles published within the last five years. The authors have made a concerted effort to select recent publications in order to present the most up-to-date discussion possible on the subject. Nonetheless, certain works—although not published within the past five years—are essential to the article. These include, for instance, books related to the theoretical framework, research methodology, global guidelines and documents relevant to the topic, as well as national and international regulations concerning professional practice, among others. Such works are primary sources and cannot be cited through secondary publications. Their inclusion in the article was deliberate, both due to their significance and because they represent the most recent theoretical, methodological, and regulatory materials originally published. However, we emphasize that, in revising the manuscript, the references were reviewed in order to comply with the request.

Reviewer 3 Report
Comments and Suggestions for Authors
The title is clear, informative and accurately reflects the content of the study. It communicates the thematic focus and the field of clinical application (pediatric nursing).
The abstract presents a clear structure (context, objective, methods, results and conclusions), using accessible scientific language. However, it is suggested to include the total number of participants; specify the qualitative method used (e.g. thematic content analysis or other designation); highlight the practical implications of the study at the end of the abstract.
The Introduction presents a good framework for the topic of preceptorship in pediatric settings and the problem is based on current literature. However, the possibility of deepening the knowledge gap that the study aims to bridge is suggested. It should also include more international references to enrich the theoretical discussion on clinical pedagogical practices in nursing.
As far as the methodology is concerned, the qualitative design is well justified; inclusion and exclusion criteria are defined; the semi-structured interview technique is well presented. However, we suggest explaining whether any reporting guidelines were used (e.g. COREQ). Include details on data analysis (systematic procedure, coding, use of software, validation between researchers); indicate whether a pilot study of the interview was carried out. It is also advisable to detail the profile of the participants (e.g. length of professional experience, specific clinical areas).
As far as the results are concerned, the emerging themes are clear, supported by direct quotes from the participants; the themes are well articulated with the objectives of the study. We suggest introducing interpretative summaries before the quotes; presenting the results with a matrix or summary table for greater visual clarity; reducing slight repetitions in the text.
The Discussion highlights the results, which are well articulated with the literature, as well as the challenges and potential of preceptorship in the pediatric context. However, it is suggested that we delve more deeply into the pedagogical implications for clinical training; reflect on the methodological limits, such as social desirability bias and the number of participants; and propose practical applications for the training of preceptors.
The conclusion is consistent with the results and highlights the relevance of preceptorship in pediatric nursing. It is suggested to avoid excessive generalizations, taking into account the qualitative method, and to reinforce suggestions for future research. Perfect ethical issues.
The quality of the writing is clear, accessible and scientifically appropriate, but we suggest a final revision by a native speaker to make minor adjustments to the flow of the language.
The references are up-to-date and relevant. However, we suggest standardizing the formatting according to the MDPI style and checking for the presence of a DOI when applicable.
Comments on the Quality of English LanguageThe quality of the writing is clear, accessible and scientifically appropriate, but we suggest a final revision by a native speaker to make minor adjustments to the flow of the language.
Author Response
Comments: The title is clear, informative and accurately reflects the content of the study. It communicates the thematic focus and the field of clinical application (pediatric nursing).
Response: Thank you for your observation.
Comments:
The abstract presents a clear structure (context, objective, methods, results and conclusions), using accessible scientific language. However, it is suggested to include the total number of participants; specify the qualitative method used (e.g. thematic content analysis or other designation); highlight the practical implications of the study at the end of the abstract.
Response:
Thank you for poiting this out. The informations requested are in the abstract already. They are indicated in red for you convenience.
Total number of participants: 14. Line 28
Method: Grounded Theory. Line 26
Implications: lines 31-34.
Comments: The Introduction presents a good framework for the topic of preceptorship in pediatric settings and the problem is based on current literature. However, the possibility of deepening the knowledge gap that the study aims to bridge is suggested. It should also include more international references to enrich the theoretical discussion on clinical pedagogical practices in nursing.
Response:
Thank you for pointing this out.
The study aims to deepen the understanding of pedagogical models that guide the practice of preceptorship in various care contexts. Given that studies on the subject indicate challenges related to the definition of concepts, insufficient institutional support, and educational models rooted in contextual specificities, the present study, by proposing to construct a theoretical model grounded in the meanings of preceptor nurses to preceptorship, aims to fill the identified knowledge gap.
The paragraphs highlighting the knowledge gap on the subject are indicated in red. Lines 65-67; 80-82.
As requested, the authors have incorporated international references on clinical pedagogical practices to enrich the theoretical discussion. Lines 59-62; 76-79.
Comments:
As far as the methodology is concerned, the qualitative design is well justified; inclusion and exclusion criteria are defined; the semi-structured interview technique is well presented. However, we suggest explaining whether any reporting guidelines were used (e.g. COREQ). Include details on data analysis (systematic procedure, coding, use of software, validation between researchers); indicate whether a pilot study of the interview was carried out. It is also advisable to detail the profile of the participants (e.g. length of professional experience, specific clinical areas).
Response:
Thank you for pointing this out.
The tool COREQ was used. Lines 103-104.
Details related to data analysis, systematic procedure, coding, use of software, validation, pilot study and profile of the participantes are indicated in red. Lines 117-122; 130-143; 158-176
Comments:
As far as the results are concerned, the emerging themes are clear, supported by direct quotes from the participants; the themes are well articulated with the objectives of the study. We suggest introducing interpretative summaries before the quotes; presenting the results with a matrix or summary table for greater visual clarity; reducing slight repetitions in the text.
Response:
Thank you for pointing this out.
The text has been revised by the authors to incorporate the suggestions. Modifications are in red.
Comments:
The Discussion highlights the results, which are well articulated with the literature, as well as the challenges and potential of preceptorship in the pediatric context. However, it is suggested that we delve more deeply into the pedagogical implications for clinical training; reflect on the methodological limits, such as social desirability bias and the number of participants; and propose practical applications for the training of preceptors.
Response:
Thank you for point this out. We agree with the comments. The requested informations are indicated in red. Lines 620-638; 593-602; 613-619; 569-572; 664-658.
Comments:
The conclusion is consistent with the results and highlights the relevance of preceptorship in pediatric nursing. It is suggested to avoid excessive generalizations, taking into account the qualitative method, and to reinforce suggestions for future research. Perfect ethical issues.
Response: Thank you for poiting this out. The informations requested are indicated in red for you convenience. Lines 664-658.
Comments:
The quality of the writing is clear, accessible and scientifically appropriate, but we suggest a final revision by a native speaker to make minor adjustments to the flow of the language.
Response: Thank you for pointing this out. The text was revised by a native speaker as requested.

Round 2
Reviewer 2 Report
Comments and Suggestions for Authors
The first revision was generally well reflected and revised.
Thank you for your hard work.
There are no further revision requests.